# Improvement in Atrioventricular Conduction Using Cardioneuroablation Performed Immediately after Pulmonary Vein Isolation

**DOI:** 10.3390/healthcare12070728

**Published:** 2024-03-27

**Authors:** Łukasz Zarębski, Piotr Futyma, Yashvi Sethia, Marian Futyma, Piotr Kułakowski

**Affiliations:** 1Medical College, University of Rzeszów, 35-959 Rzeszów, Poland; lukasz.zarebski@interia.pl (Ł.Z.); sethiayashvi01@gmail.com (Y.S.); 2St. Joseph’s Heart Rhythm Center, 35-623 Rzeszów, Poland; marian.futyma@wp.pl (M.F.); piotr.kulakowskimd@gmail.com (P.K.); 3Department of Cardiology, Postgraduate Medical School, Grochowski Hospital, 04-073 Warsaw, Poland

**Keywords:** cardioneuroablation, atrial fibrillation, pulmonary vein isolation, atrioventricular block

## Abstract

In patients with atrial fibrillation (AF) recurrences after pulmonary vein isolation (PVI), concomitant treatment using anti arrhythmic drugs (AADs) can lead to clinical success. However, patients with atrioventricular (AV) block may not be good candidates for concomitant AAD therapy due to the risk of further worsening of conduction abnormalities. Cardioneuroablation (CNA), as an adjunct to PVI, may offer a solution to this problem. We present a case of a 74-year-old male with paroxysmal AF and first degree AV block in whom CNA following PVI led to PR normalization. The presented case describes an example of CNA utilization in patients with AF undergoing PVI who have concomitant problems with AV conduction and shows that CNA can be sometimes useful in older patients with functional AV block.

## 1. Introduction

Pulmonary vein isolation (PVI) is a cornerstone of atrial fibrillation (AF) ablation; however, in some subgroups, the recurrence rate still remains substantial [1]. In patients with AF recurrences after PVI, a concomitant treatment using antiarrhythmic drugs (AADs) can lead to clinical success [2]. Nevertheless, patients with atrioventricular (AV) block may not be good candidates for accompanying AAD therapy due to the risk of further worsening of conduction abnormalities. Drugs such as propafenone, amiodarone or β-blockers may prolong the PR interval even more, potentially leading to further disruptions in the electrical conduction system of the heart. For patients with AV block, organic or functional, a consideration of pacemaker implantation emerges as a potential intervention. However, unlike the organic AV block, which is frequently related to structural heart diseases, the functional AV block can be related to physiological factors and is sometimes reversible. Therefore, in cases where AV block is functional rather than organic, alternative strategies may warrant exploration to avoid the potential complications associated with permanent pacing such as lead malfunction, pacemaker dependency and right ventricular pacing-induced cardiomyopathy.

Cardioneuroablation (CNA), as an adjunct to PVI, may offer a solution to this problem [3,4]. CNA involves radiofrequency (RF) energy to ablate ganglionated plexi (GP) situated within the epicardial fat surrounding the atria. The RF current affects GP and subsequently reduces vagal tone due to the prevalence of parasympathetic fibers within GP. This technique has been recently introduced into clinical practice to treat reflex syncope due to functional sinus arrest or AV block [5,6,7,8]. The two-year efficacy of CNA ranges between 80 and 90% in patients with syncope. The use of CNA in patients undergoing ablation is less well established [9,10,11]. Moreover, it remains unknown if patients with AF and coexisting functional AV block can be candidates for PVI and concomitant CNA performed during the same ablation session.

## 2. Case Report

A 74-year-old male with paroxysmal AF, ineffectively controlled with β-blockers and propafenone, was admitted for elective PV re-isolation. His first PVI took place in 2003 and, later, he underwent cavotricuspid isthmus ablation for typical atrial flutter in 2008. The patient complained of recurrent palpitations once a week for several years. Apart from AF, the patient had a history of first-degree AV block, which made AAD up-titration problematic. His medication consisted of acenocumarol 3 mg daily under INR control, bisoprolol 5 mg daily and propafenone 300 mg twice a day. His only comorbidity was hyperlipidemia. A 12-lead ECG showed a normal sinus rhythm with first-degree AV block. Echocardiography showed second degree mitral regurgitation (small), and second degree tricuspid regurgitation (small). Slight dilatation of both atria and the right ventricle was observed. The left ventricular ejection fraction was 60%. The patient was provided with comprehensive information about the procedure, potential risks, benefits, and alternative treatments. Afterward, the patient agreed to undergo the ablation, signing a consent form. Immediately before the procedure, a PR interval of 444 ms was measured using the calipers of an electrophysiological system (EP Tracer, Cardiotek, Maastricht, The Netherlands) (Figure 1). The ultra-short-term deceleration capacity was also measured right before the procedure and its value was 7.4 ms, suggestive of increased parasympathetic activity [12]. Under ultrasound guidance, two vascular sheaths were introduced to the right femoral vein. Next, a single transseptal puncture was performed and the sheath for contrast injection was positioned in the left atrium. An ablation catheter (AC) was introduced into the right ventricle and, during rapid right ventricular pacing from the AC, a rotational scan was performed for left atrial reconstruction with the EP Navigator (Philips Medical Systems, Best, The Netherlands) 3D mapping system. The ablation catheter was introduced into the left atrium, which was subsequently mapped, and PV ostia were determined. Mapping showed the reconnection of all four PVs so, given a paroxysmal form of AF, the decision was made to proceed with PV re-isolation as an endpoint for AF ablation. Using a high-power short-duration technique (50 W, 8–15 s per RF application), numerous circumferential applications around the PV ostia were performed and the re-isolation of all four PVs was successfully achieved. PVI was further confirmed by a lack of capture during high-output pacing maneuvers performed at ablation lines and from the inside of all four PVs.

After successful PV re-isolation, no change in the PR interval was observed. Using an anatomical approach, the ablation catheter was positioned at the location of the presumed area of the inferior para-septal ganglionated plexus (IPSGP), which mainly controls the function of AV node, where fragmented signals, suggestive of IPSGP, were recorded (Figure 2). An additional radiofrequency application in the vicinity of the IPSGP was performed (13 s, 56 W), which led to almost immediate PR shortening (Figure 3). During the waiting time of 10 min, no PR interval prolongation was observed (Figure 4). The ultra-short-term deceleration capacity was measured again and its value was 4.25 ms after CNA. The patient remained symptom-free and was successfully discharged on the next day. All AADs were discontinued.

During the blanking period, one week after the procedure, the patient was admitted to our outpatient clinic and the recurrence of AF was recorded. Thus, amiodarone 200 mg was prescribed, which subsequently led to the successful restoration of the sinus rhythm. No PR interval prolongation occurred. Then, 24 h Holter monitoring was performed one month after the procedure. The average heart rate was 59. No atrial fibrillation or AV block of any grade were detected. During the 3.5 month follow-up, the patient remained symptom-free.

## 3. Discussion

There are several unresolved issues concerning CNA. A variety of interactions between autonomic functions and AF incidence have been investigated throughout recent decades [13]. Most importantly, an increased activity of parasympathetic tone can have a significant impact on proper AF management. In the era of PVI as a preferred strategy for maintaining sinus rhythm, more attention is given to the possible impact of the ablation around the PV ostia on parasympathetic activity. The very close proximity of GPs near the ablation lesion set required for effective PVI frequently makes a parasympathetic attenuation an inevitable component of an effective AF ablation procedure [14]. Tools and techniques for PVI are divergent and a lesion set can differ individually between subjects, mainly because of anatomical differences. Given this, the level of PVI influence on GP can differ, and some patients who can possibly benefit from an additional PVI impact on autonomic function may need an additional ablation lesion set closer to the suspected GP sites.

Nevertheless, even though we have already gained a more comprehensive understanding of the anatomy and physiology of the main atrial GPs, there are still several unresolved issues concerning CNA. For example, determining the optimal GP location may remain a challenge and the most effective approach to this issue is currently unclear. In our patient, we used an anatomical approach, taking advantage of rotational angiography, which creates an accurate right atrial and left atrial anatomy. In this way, a triangular area between the right and left atrium, where inferior paraseptal GPs are located, could be easily identified and ablated. Other GPs location methods may currently include a spectral or time-domain analysis of fragmented potentials, computed tomography, intracardiac echocardiography or the identification of GPs through high-frequency stimulation (HFS) to induce vagal responses. These methods add complexity to the procedure, may require additional equipment and some of them, like HFS, may require general anesthesia. In our patient, we used a simple method which turned out to be sufficient and resulted in a successful procedure. Another important issue regarding CNA is establishing an appropriate procedural endpoints, as there is still uncertainty about how to identify complete vagolysis and whether incomplete ablation may impact the success rate [15]. Previous studies have employed two methods for defining endpoints: through the elimination of all targeted electrograms and through the electrophysiological evidence of vagal denervation, detectable through changes in heart rate or AV conduction properties [13]. An additional approach involves using atropine response as an endpoint, where a positive response before the procedure, that becomes negative afterward, indicates a successful outcome. Moreover, extracardiac vagal stimulation, performed with an electrophysiological catheter placed in the internal jugular vein, may be promising option [16]. While there is no precisely defined extent of vagolytic endpoint necessary for achieving a clinical response, the utilization of these quantifiable parameters provides a basis for assessing the efficacy of the procedure [17]. In our patient, we used the deceleration capacity as a marker of parasympathetic activity, which also indicated a decrease in vagal tone after CNA (7.4 vs. 4.25 ms). Such elevated deceleration capacity measurements at baseline were suggestive of increased vagal tone, and such increased activity likely contributed to the functional form of AV block in the presented patient.

The extrinsic cardiac autonomic nervous system regulates the function of the sinoatrial (SA) and AV nodes. Studies have shown that GPs have numerous interconnections between themselves to modulate sinus and AV nodal function [18]. It was suggested that GPs situated away from the SA and AV nodes engage in interactions with those in closer proximity to these nodes through interconnections within the atrial neural network, which constitutes the intrinsic cardiac autonomic nervous system. Consequently, GPs positioned at a distance from the SA and AV nodes have the capacity to influence electrophysiological characteristics, including the sinus rate and AV conduction. However, the IPSGP is thought to control AV nodal function first of all [19,20]. This assumption is based on observations that stimulating this GP resulted in AV block and a reduction in the effective refractory period (ERP) of neighboring atrial tissue, while the sinus rate remained unaffected [19]. Unlike this, the GP located in the anterior right region predominantly influences the SA node function and the electrophysiological characteristics of the surrounding atrial tissue [20,21]. Stimulating the parasympathetic nerves near the superior paraseptal ganglionated plexus (SPSGP) resulted in a reduction in the sinus rate and a shortened ERP of the neighboring atrial tissue, with no significant impact on AV conduction [20,21]. This is the reason why CNA focused on AV block treatment should be targeted on IPSGP rather than on SPSGP. In our patient, we chose to ablate IPSGP because there was a problem with AV conduction and the sinus node automaticity was proper in this case. If such an approach had not been effective, other GPs located close to the left atrium might have been targeted, including the left inferior GP, area of ligament of Marshall, as well as coronary sinus. In our patient, this complex approach was not necessary because a single RF application at the area of IPSGP proved effective. The normalization of the PR interval following CNA in the context of AF holds significant clinical relevance and potential benefits for patients. Recent studies demonstrated a notable correlation between PR interval prolongation and a heightened frequency of AF occurrence. Several potential explanations were proposed for understanding the mechanism of this correlation [22]. Primarily, an elongated PR interval serves as an important electrocardiographic indicator that warrants careful consideration, as it may signify the presence of underlying heart pathology. Notably, this prolonged interval could be indicative of various cardiovascular conditions, including myocardial ischemia or decompensated heart failure, which both contribute to structural and electrical remodeling, causing atrial inflammation and fibrosis [23,24]. These conditions can lead to irregular conduction delays, creating a substrate for the perpetuation of AF and making patients more susceptible to arrhythmia recurrences. Secondly, predisposition towards hypervagotonia contributes to an increased burden of AF. In previous studies, patients with prolonged PR interval had lower baseline heart rates [22]. Bradycardia is well known for creating a favorable environment to initiate and sustain AF episodes. A low heart rate may also lead to a prolonged atrial refractory period, making the atria more susceptible to premature atrial contractions or short atrial tachycardia runs, which can facilitate the onset of AF. This phenomenon is particularly relevant in understanding the mechanisms behind AF occurrences, especially in individuals exhibiting enhanced parasympathetic tone. Finally, genetic correlations exist between the PR interval and AV node conduction [25]. A recent study revealed 44 loci which contain genes that are over-represented in cardiac disease processes, including heart block and atrial fibrillation [26]. A genetic predisposition to prolonged PR interval signifies a considerable remodeling of the sinus node, atrial substrate, AV node, and His bundle branches.

Apart from the impact of GPs on AV conduction, it is worth mentioning their overall influence on AF. Numerous clinical investigations have explored the impact of GP ablation on AF burden, either as a standalone procedure or as an adjunct to PVI [27,28,29]. This approach acknowledges the heterogeneity of AF mechanisms, recognizing that targeting both electrical triggers and the fact that autonomic modulation is crucial for the effective treatment of AF. An examination of comparative studies featuring PVI alone, GP ablation alone or their combined procedures has revealed several interesting findings. Stand-alone PVI has demonstrated higher success rates compared to GP ablation procedures alone [27]. However, when contrasting PVI with combined PVI + GP ablation procedures, success rates exhibit a significant increase [28,29,30]. The success rates of PVI + GP ablation can vary widely; however, these are higher for patients with paroxysmal AF (between 70% and 94%) [28,30] than those with long-standing persistent AF (49%) [29]. It is important to take into account the potential denervation during PVI, as this procedure has the potential to inadvertently disrupt neural signalling pathways and alter the autonomic innervation of the heart. Parasympathetic GPs are typically situated in close proximity to the ostia of pulmonary veins [13]—the main target of PVI—and can be frequently modified during the AF ablation procedure. The application of RF energy during PVI, crucial for achieving the electrical isolation of the pulmonary veins, can impact adjacent neural structures, such as GP. Such unintentional cardioneurmodulation cannot replace a complete CNA procedure; however, it is crucial to remember that such effects occur and may have significant importance. Recent experimental studies showed that intentional CNA can predispose future malignant ventricular arrhythmias in animal models [31]. Thus, more long-term data on the exact impact of CNA regarding any possible future occurrence of adverse events related to potentially malignant ventricular arrhythmias would be welcomed. While some reassuring information on the long-term safety of CNA can be projected from several large, multi-center, randomized controlled trials on the ablation of AF using thermal PVI, more information is needed on this topic, especially in some subpopulations with increased parasympathetic activity.

## 4. Conclusions

The presented case describes an example of CNA utilization in patients with AF undergoing PVI who have concomitant problems with AV conduction. It shows that CNA can be useful not only in young but sometimes also in older patients with functional paroxysmal or persistent AV block [5,6,7]. In such patients, additional ablation at the area of inferior GPs, which are responsible for the autonomic control of AV conduction, may be effective. Such an approach may obviate the need for pacemaker implantation, enable the safe use of AAD and perhaps increase quality of life without the risk of the further progression of the AV conduction block. PR interval normalization following CNA offers an alternative and promising approach in such cases. By addressing autonomic control through CNA, the procedure restores the normal functioning of the atrioventricular node and gives an opportunity to provide safe AAD therapy. Diagnostic methods documenting a functional rather than organic AV block, such as a tilt-test, atropine testing or EP study excluding intra- or infra-hisian block (unfortunately not carried out in the presented case), should be sought to appropriately identify those candidates who can benefit from additional CNA during PVI. When identifying potential candidates with AV block to can, it is necessary to remember that this method only applies to patients with functional AV block. Organic disorders, such as myocardial ischemia, heart failure or the genetic remodeling of an AV node, cannot be taken into consideration. Also, this procedure is usually successful for patients with functional sinus bradycardia with concomitant AF in whom CNA is confined to the area of SPSGPs [8].

## Figures and Tables

**Figure 1 healthcare-12-00728-f001:**
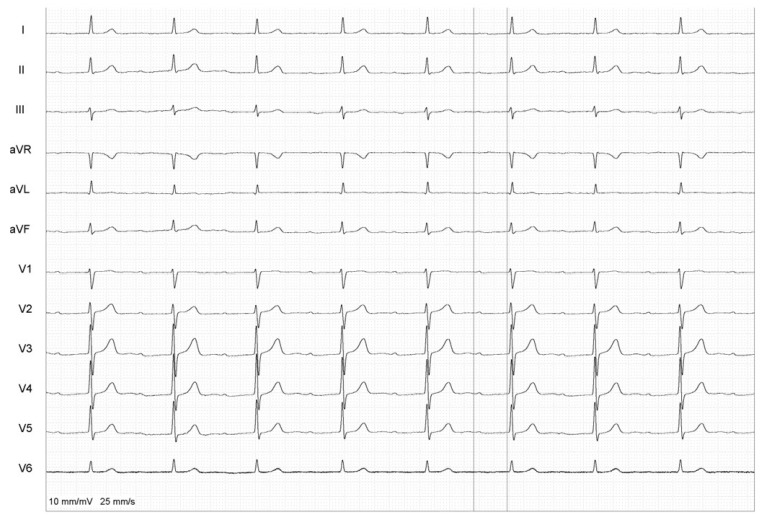
Baseline ECG.

**Figure 2 healthcare-12-00728-f002:**
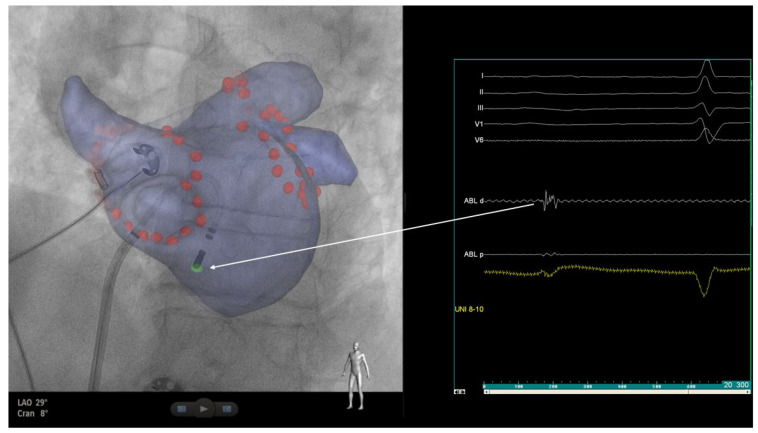
Fluoroscopic image in LAO projection with 3D left atrial map showing an ablation catheter (AC) positioned at the location of the inferior para-septal ganglionated plexus (IPSGP). Fragmented signals were recorded from the distal pair of AC electrodes at the IPSGP area.

**Figure 3 healthcare-12-00728-f003:**
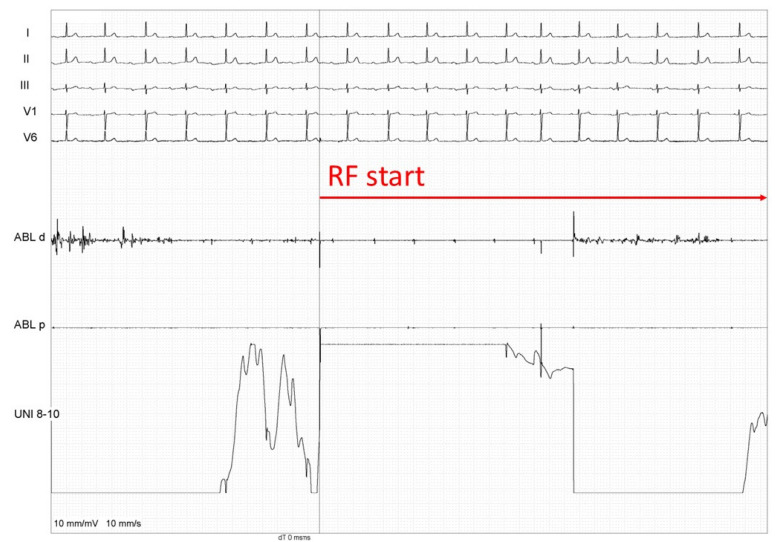
PR shortening during radiofrequency (RF) ablation at the IPSGP area.

**Figure 4 healthcare-12-00728-f004:**
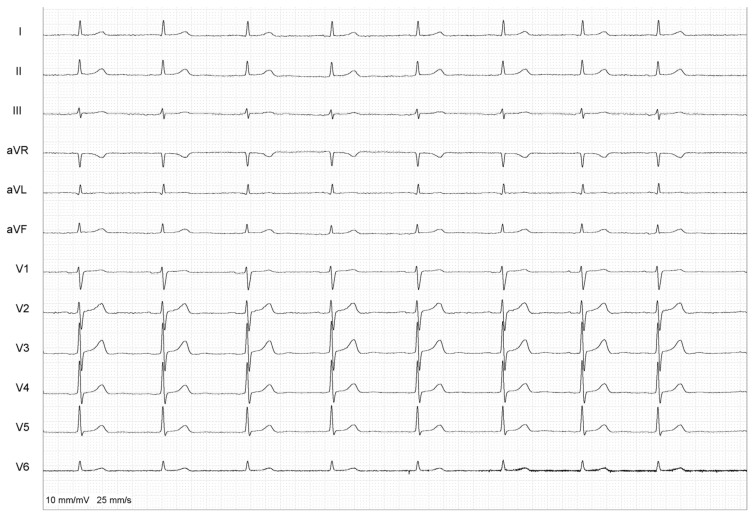
Final ECG after PVI and CNA of IPSGP.

## Data Availability

Data are contained within the article.

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
