# Peer review of "Improvement in Atrioventricular Conduction Using Cardioneuroablation Performed Immediately after Pulmonary Vein Isolation"

_healthcare, 2024, doi:10.3390/healthcare12070728_

Round 1
Reviewer 1 Report
Comments and Suggestions for Authors
Sethia, Kulakowski et al present a fascinating case on an emerging topic that demonstrates the possibility of effectively modulating (intentionally and targeted, not just coincidentally) the vagal tone on the atrioventricular node simultaneously with PVI, thereby allowing antiarrhythmic therapy optimization without the need for a pacemaker, and discuss the current scientific evidence and clinical implications. Here are my comments:
Line 31: Please define briefly the difference between organic and functional AV block.
Line 42: Was there any non cardiologic history?
Line 53: I think it should be clarified how the differential diagnosis between functional and organic AVB was established for this patient (considering the age, as well). Was an atropine test conducted, or was a Tilt-test (regardless of syncope history) conducted to search for cardioinhibitory/vasodepressor responses? Was an EP study performed to exclude intra- or infra-hisian block?
Line 53-127: Please explain the rationale of using only the ultrashort deceleration time as an endpoint, and why the the value of 7.4ms was considered sufficient as a cutoff for high vagal tone.
Line 57: Which mapping system did you use?
Line 61: Since this was a Re-Do in a patient who already underwent two procedures, was Box ablation considered? If yes, why it was excluded? Please explain it to the reader.
Please report on the process of informed consent acquisition.
Line 70: Please provide a brief explanation, contextualized for the patient, on the reason for ablating only the IPSGP (as further detailed in the discussion section).
Line 71: How many watts and for how many seconds were applied?
Line 86: Please specify how many hours the Holter ECG was.
146-151: Please separate the section describing the case from the section discussing the theoretical and clinical implications.
Author Response
Line 31: Please define briefly the difference between organic and functional AV block.
Thank you for this comment. We added:
- Page 1, Line 31: “However, unlike the organic AV block, which is frequently related to structural heart diseases, the functional AV block can be related to physiological factors and is sometimes reversible.”
Line 42: Was there any non cardiologic history?
Thank you for this comment. We added:
- Page 2, Line 54: “His only comorbidity was hyperlipidemia.”
Line 53: I think it should be clarified how the differential diagnosis between functional and organic AVB was established for this patient (considering the age, as well). Was an atropine test conducted, or was a Tilt-test (regardless of syncope history) conducted to search for cardioinhibitory/vasodepressor responses? Was an EP study performed to exclude intra- or infra-hisian block?
Thank you for this comment. Unfortunately, neither atropine test nor tilt test nor EP study excluding intra- or infra-hisian block were not performed in our study. We added this information to the manuscript:
- Page 7, Line 234: “Diagnostic methods documenting functional rather than organic AV block such as tilt-test, atropine testing or EP study excluding intra- or infra-hisian block (unfortunately not done in the presented case) should be sought to appropriately identify those candidates who can benefit from additional CNA during PVI.”
Line 53-127: Please explain the rationale of using only the ultrashort deceleration time as an endpoint, and why the value of 7.4ms was considered sufficient as a cutoff for high vagal tone.
Thank you for this comment. We relied on the literature data, where deceleration capacity > 7.12ms discriminate patients with high vagal tone and vasovagal character of the syncope. We added this reference to the manuscript:
- Page 8, Line 280: “Zheng, Lihui et al. “The Diagnostic Value of Cardiac Deceleration Capacity in Vasovagal Syncope.” Circulation. Arrhythmia and electrophysiology vol. 13,12 (2020): e008659.”
Line 57: Which mapping system did you use?
We modified: “…left atrial reconstruction with the EP Navigator (Philips Medical Systems, Best, The Netherlands) 3D mapping system.”
Line 61: Since this was a Re-Do in a patient who already underwent two procedures, was Box ablation considered? If yes, why it was excluded? Please explain it to the reader.
Thank you for this comment. The patient underwent two procedures previously: PVI for AF and CTI ablation for typical atrial flutter. Given the fact that the AF was still paroxysmal, not persistent, the decision was to proceed with re-PVI only. We modified:
- Page 2, Line 72: “Mapping showed reconnection of all four PVs so, given a paroxysmal form of AF, the decision was made to proceed with PV reisolation as an endpoint for AF ablation.”
Please report on the process of informed consent acquisition.
Thank you for this comment. We added:
- Page 2, Line 58: “The patient was provided with comprehensive information about the procedure, potential risks, benefits, and alternative treatment. Afterward, the patient consciously agreed to undergo the ablation, signing a consent form.”
Line 70: Please provide a brief explanation, contextualized for the patient, on the reason for ablating only the IPSGP (as further detailed in the discussion section).
Thank you for this comment. We added:
- Page 3, Line 83: “…which mainly control the function of AV node”
Line 71: How many watts and for how many seconds were applied?
Thank you for this comment. We added:
- Page 3, Line 86: “(13 seconds, 56 W)”
Line 86: Please specify how many hours the Holter ECG was.
Thank you for this comment. We added:
- Page 4, Line 100: “24-hour Holter monitoring…”
146-151: Please separate the section describing the case from the section discussing the theoretical and clinical implications.
The "Case Report" section is separated from the "Discussion" section: both these sections are separate paragraphs (2 & 3 respectively).
Reviewer 2 Report
Comments and Suggestions for Authors
I had the opportunity to review your case report and wanted to commend you on the thorough research and clarity of presentation. It's certainly a valuable contribution.
However, considering the specific focus of your case, I believe it might find a more suitable audience in a journal specializing in Electrophysiology (EP). An EP-focused journal could provide the right context and readership that aligns closely with the subject matter of your report.
Author Response
Thank you for your suggestion. We acknowledge the fact that our manuscript deals with electrophysiological area which is trending recently. However, we believe that providing our observations to a broader audience can increase awareness that some patients, even elderly ones, can benefit from atrial fibrillation ablation combined with such cardioneuroablative effects. These effects can have an impact on decision process regarding therapeutic considerations for pacemaker implantation for AV block vs catheter ablation for AF.
Reviewer 3 Report
Comments and Suggestions for Authors
The case-report is well written and organized and has a substantial increment to the subject.
However, I have the below comments to improve the manuscripts.
1. Since this case report is more about cardioneuroablation, it would be better if the authors discuss/explain more about it in the introduction section. Especially elucidating about the anatomical, physiological, electrophysiological principles behind it and its relevance to the case discussed.
2. Line 42, “B-blockers” to be corrected as β- blockers or beta blockers.
3.Relevant subheadings can be included for a more crisp presentation of this case report. For instance from lines 68 – 76, the authors try to explain/ determine whether the block is functional or not via the Holter tracings. This could be outlined as “determining whether the block is functional”.
4.Similarly, the authors could validate the suitability of the case for cardioneuroablation
Comments on the Quality of English Language
The manuscript MUST be revised for English language (grammar, syntax and spelling)
Author Response
Since this case report is more about cardioneuroablation, it would be better if the authors discuss/explain more about it in the introduction section. Especially elucidating about the anatomical, physiological, electrophysiological principles behind it and its relevance to the case discussed.
Thank you for this comment. We added:
- Page 1, Line 38: “CNA involves radiofrequency (RF) energy to ablate ganglionated plexi (GP) situated within the epicardial fat surrounding the atria. The RF current affects GP and subsequently reduces vagal tone due to the prevalence of parasympathetic fibers within GP.”
- Line 42, “B-blockers” to be corrected as β- blockers or beta blockers.
Thank you for this comment. We corrected it.
3.Relevant subheadings can be included for a more crisp presentation of this case report. For instance from lines 68 – 76, the authors try to explain/ determine whether the block is functional or not via the Holter tracings. This could be outlined as “determining whether the block is functional”.
4.Similarly, the authors could validate the suitability of the case for cardioneuroablation
Thank you for these suggestions. We agree that more detailed subheadings could be somehow beneficial for more detailed digesting the topic of functional AV blocks. However, given the initial aim of submission (brief case report), in our opinion, this could actually lead to further overwriting. We further elaborated the topic of functional AV block determination in our patient:
- Page 5, Line 144: “Such elevated deceleration capacity measurements at baseline were suggestive for increased vagal tone, and such increased activity likely contributed to functional form of AV block in the presented patient.”